# Recent Insights from Molecular Dynamics Simulations for G Protein-Coupled Receptor Drug Discovery

**DOI:** 10.3390/ijms20174237

**Published:** 2019-08-29

**Authors:** Ye Zou, John Ewalt, Ho-Leung Ng

**Affiliations:** Department of Biochemistry and Molecular Biophysics, Kansas State University, Manhattan, KS 66506, USA

**Keywords:** GPCRs, membrane protein, molecular dynamics, protein structure, drug design, biased-signaling pathway, allosteric sites

## Abstract

G protein-coupled receptors (GPCRs) are critical drug targets. GPCRs convey signals from the extracellular to the intracellular environment through G proteins. Some ligands that bind to GPCRs activate different downstream signaling pathways. G protein activation, or β-arrestin biased signaling, involves ligands binding to receptors and stabilizing conformations that trigger a specific pathway. β-arrestin biased signaling has become a hot target for structure-based drug discovery. However, challenges include that there are few crystal structures available in the Protein Data Bank and that GPCRs are highly dynamic. Hence, molecular dynamics (MD) simulations are especially valuable for obtaining detailed mechanistic information, including identification of allosteric sites and understanding modulators’ interactions with receptors and ligands. Here, we highlight recent MD simulation studies and enhanced sampling methods used to study biased G protein-coupled receptor signaling and their conformational dynamics as well as applications to drug discovery.

## 1. Introduction

G protein-coupled receptors (GPCRs) are the largest superfamily of membrane proteins in the human genome [1]. They are also the most important and largest collection of pharmacological drug targets [2]. These receptors share modest sequence similarity but high structural conservation, all with seven transmembrane-spanning helices, an extracellular N terminus, and an intracellular C terminus. The crystal structure of bovine rhodopsin, one of the most studied GPCRs, shows the seven helices forming a helical cylinder, which is linked by three intracellular and three extracellular loops [3]. There are five main GPCR families: rhodopsin (class A), secretin (class B), glutamate (class C), frizzled/taste (class F), and adhesion [4]. GPCRs’ typical functions are the translation of extracellular stimulation into intracellular signals via the binding of different ligands, which then cause different conformational changes and downstream effects. Each receptor can activate specific G proteins and regulate unique downstream signals. The GPCR ligands bind to these receptors and stabilize conformations, then regulate and modulate various intracellular transduction processes. GPCR agonist ligands are extremely diverse and include photons, ions, odorants, tastants, small molecule neurotransmitters, amino acids, polypeptides, hormones, nucleotides, and lipids [5,6]. Classical GPCR activation involves an agonist-induced conformational change. This causes the receptor to interact with the Gα subunit, part of the heterotrimeric GTP-binding proteins (G proteins), which then dissociates from the Gβγ subunits and binds to guanosine diphosphate (GDP). Research has also shown that there are also at least five different activation modes different from classical activations [7,8]. These are involved in phenomena such as intracellular activation, dimerization activation, transactivation, biphasic activation, and biased activation. Because GPCRs are involved in these activations, which are related to many human diseases, they are common drug targets [9,10,11,12]. GPCRs are normally activated on the cell surface. GPCRs can also be activated from inside the cell, which is called intracellular activation [13]. GPCR activation functions also depend on the forms of the GPCRs, monomeric or dimeric, the latter of which is called dimerization activation [14]. GPCRs can be activated by ligands and these activated GPCRs can transactivate other proteins, such as receptor tyrosine kinases (RTKs), which then can activate Ras/MAP kinases. This is called transactivation [15]. GPCRs can also activate two different phases of signaling. This is called biphasic activation [16]. The last activation is called biased activation (Figure 1). The biased activation (also called biased signaling pathway) involves parallel G protein-independent signaling pathways. Instead of activating G proteins, there are activating β-arrestins, which mediate downstream signaling. The main functions are internalization and desensitization. Activated by agonists, the GPCRs are phosphorylated by GPCR kinases (GRKs) on multiple sites of the C-terminus; arrestins will bind to these phosphorylated sites, and G protein-coupling will be blocked by the arrestin-GPCR complex. This is called biased activation [4,17,18,19,20]. In this review article, we focus on the biased-signaling pathway and β-arrestins. There are four classes of arrestins: arrestin 1 (visual arrestin), arrestin 2 (β-arrestin 1), arrestin 3 (β-arrestin-2), and arrestin 4 (cone arrestin). Arrestins are highly conserved, with approximately 50% sequence homology between vertebrates and invertebrate primary structures [21,22,23,24,25]. Because the functions of arrestins are diverse, β-arrestin biased-signaling pathways are of significant pharmaceutical interest. The current understanding of the structural conformations related to biased signaling is sparse.

Recent studies have revealed that GPCRs are dynamic proteins with multiple conformational changes depending on ligand binding, signaling proteins, and the membrane environment [26,27]. Several crystals of class A GPCRs in the active state show there are conformational changes in the intracellular domain and transmembrane helices 5, 6, and 7 (TM5, TM6, and TM7) of the receptors [28,29,30]. Biophysical research has shown that the binding affinity to an agonist is increased by coupling the G protein to the receptor [31]. After a ligand binds to a receptor, it causes and stabilizes conformational changes [32]. Agonists binding to GPCRs induce and mediate different downstream pathways, either through G protein activation or β-arrestin biased signaling. Typically, the primary ligand binding sites, orthosteric sites, are highly conserved. This presents a significant challenge for drug discovery. Ideally, a designed ligand has high selectivity and only activates specific receptors; an orthosteric site that is highly conserved between related receptors increases the difficulty of doing so. Because of the challenges these orthosteric sites present, researchers have been trying to explore new alternative binding sites, which could increase both binding affinity and decrease off-target effects. Allosteric binding sites share less homology compared with orthosteric sites, so they have become increasingly attractive to researchers [33]. Positive allosteric modulators can increase the potency of orthosteric ligands; negative allosteric modulators can decrease the potency of the response to orthosteric ligands [34]. Several crystal structures of GPCRs bound with allosteric modulators have already been solved. Kruse et al. solved a crystal structure of a GPCR (M2 muscarinic receptor) with an allosteric modulator [30]. Dore et al. solved a class C GPCR (metabotropic glutamate receptor 5) bound with an allosteric modulator [35]. Jazayeri et al. solved a class B GPCR (glucagon receptor) bound with an allosteric antagonist [36]. However, it is also challenging to determine crystal structures of GPCRs with allosteric modulators, and few are in the Protein Data Bank (PDB). Therefore, computational methods are valuable in helping to identify new allosteric binding sites and offer new structural information of GPCRs and their interactions with ligands.

Molecular dynamics simulations are important computational methods widely used in many fields of study. Simulations assist researchers in obtaining structural information, specifically, the conformational states that are hard to capture by experimental methods. Because of rapid technological development, computing is easier and faster than ever. This has also allowed researchers to run long time scale simulations to obtain more detailed mechanistic information [37]. In this review, we will discuss the insights on GPCR interactions with ligands revealed by molecular dynamics simulations and enhanced sampling techniques.

## 2. New Insights from Molecular Dynamics Simulations

Computational and simulation methods provide other ways to explore complex systems that are difficult to study with current experimental methods. Simulation via molecular dynamics (MD) is a very powerful and easy to use computational tool. It can help researchers determine receptor-ligand structures, dynamics of binding, and binding kinetics and functions [38,39,40,41]. More recently, MD simulations have been used to study macromolecular conformational dynamics on longer time scales, up to a millisecond and beyond [42,43,44,45]. Computational and simulation methods can play useful roles in drug discovery as well. These methods have helped improve our understanding of GPCRs’ structures and functions [2,46,47,48,49]. There are three main requirements for an MD simulation: the model system, the force field, and the MD simulation software. The most popular MD simulation packages include AMBER [50], CHARMM [51], GROMACS [52], and NAMD [53]. These packages are making simulations easier to perform and are quick to adopt new technological and methodological advances.

### 2.1. Using Molecular Dynamics Simulations to Study GPCR-Ligand Binding

Biased signaling generates functional selectivity and has attracted notable drug discovery interest. The structures and mechanisms involved in biased signaling are still not clear as there are very few GPCR crystal structures available.

The μ-opioid receptor (μOR or MOR) is the first GPCR that demonstrated β-arrestin-biased signaling [54]. The μOR is the primary target for strong analgesics [55]. The best-known opioid agonists are opiate drugs, which are among the oldest medicines and are analgesics [56]. Even though opiates are widely used, they have notorious side effects including addiction, respiratory suppression, and constipation. β-arrestins act as negative regulators in the μOR signaling pathways [57,58]. Recent research has supported a trend in which ligands that are more biased towards the β-arrestin pathway are associated with increased undesired side effects [59,60]. Currently, there are two μOR crystal structures and two electron microscopy structures available in the Protein Data Bank (PDB: 4DKL, 5C1M, 6DDE/6DDF) [29,61,62]. These high-resolution structures offer the possibilities of using these structures to perform simulations and potentially assist in the discovery of novel drugs with fewer side effects [9,63,64].

Crystallographic studies of μOR bound with the potent agonist, fentanyl, involved active and inactive states. The crystal structures used are PDB 5C1M and 4DKL. 4DKL is the structure of inactive μOR with the irreversible morphinan antagonist β-funaltrexamine (β-FNA), which can be seen in Figure 2. 5C1M is the structure of active μOR with the agonist BU72. The primary structural difference between active and inactive states is TM6 shifting outwards by 10.3 Å [29].

Fentanyl is an analgesic that is much more potent and dangerous than morphine [65]. Unlike morphine, fentanyl can strongly induce β-arrestin biased signaling [66]. Both morphine and fentanyl have a protonatable tertiary amine in the piperidine ring. Compared to morphine, fentanyl is more flexible (Figure 3). Lipinski et al., using the 4DKL and 5C1M crystal structures and manually docking morphine and fentanyl into the protein, found that the mutation of Ser329^7.46^ to alanine, located in the sodium binding pocket, is sensitive to the N-phenethyl chain of fentanyl [67]; mutations involving Trp318^7.35^ and His319^7.36^ to methionine demonstrate similar sensitivities to the N-phenethyl chain of fentanyl [68]. The superscript decimal numbers seen in the previous sentence refer to the Ballesteros–Weinstein numbering scheme, with the number to the left of the decimal referring to which of the seven transmembrane helices the residue is on and the number to the right of the decimal giving the relative position to the most conserved residue on the helix, which is numbered 50 [69].

In the fentanyl binding mode, both active and inactive receptors are stimulated. In the inactive mode, there is a sodium cation in an allosteric site. The simulation results of active and inactive binding interactions are similar. The two ligands both have the amine of the piperidine ring protonated. In the binding mode, the protonated amines interact with residue Asp147^3.32^; hence, the N-phenethyl chain is facing the intracellular side. The piperidine ring plays an important role in this binding mode. It interacts not only with A147^3.32^ but also with Gln124^2.60^. The N-phenethyl chain forms hydrophobic contacts with Try326^7.43^, Ile296^6.51^, Ile322^7.39^, and Trp293^6.48^. In the morphine binding mode, the simulation results of active and inactive binding interactions are also similar. The protonated amine in morphine also interacts with residue Asp147^3.32^. Morphine’s binding pocket is similar to fentanyl’s. The protonated amine is facing intracellularly. The phenol and the ether group are facing the extracellular side and are exposed to the solvent. The CHARMM-GUI service [70] was used to parameterize and prepare the sample for simulation. The receptor was placed in a phosphatidylcholine (POPC) membrane and solvated with the TIP3P water model; the CHARMM36 force-field was used. The simulation runtime was 1.2 μs [71].

In this research, MD simulations were also used to study a “Trp rotamer toggle switch”, which acts as a transmission switch. Trp293^6.48^ was found to be a highly conserved residue, and multiple MD simulations support its central role in conformational change [72,73,74]. Analysis of MD simulations monitored the movement of Trp293^6.48^ and focused on two different dihedral angles, χ_1_ and χ_2_. Three rotamers were observed that differed from those seen in the crystal structure. Such rotamers would have been difficult to observe by experimental means.

### 2.2. Using Molecular Dynamics to Predict Arrestin-Biased Ligands

Pursuing biased signaling is an alternative strategy to discover highly selective drugs [75]. However, a detailed understanding of the biased signaling pathway is still incomplete. In particular, the crystal structures reported are of the active and inactive states, with very few structures that are relevant to intermediate conformations. MD simulations have assisted in revealing these intermediate conformations and the mechanisms of biased signaling. We discuss an example that used MD simulations to predict arrestin-biased ligands.

Recently, McCorvy et al. reported using the D2 dopamine receptor (D2R) as a model to study GPCR-ligand binding, which involves biased signaling, by MD simulations [76]. D2R is a primary drug target for schizophrenia and Parkinson’s disease. The clinical implications of differently biased D2R ligands is a topic of great research interest. The authors describe a new method to use MD simulations to design β-arrestin biased ligands. The crystal structure of the complex of β2 adrenergic receptor (β2AR) bound to epinephrine shows that the ligand forms a hydrogen bond network with the conserved TM5 serine residues [77]. Three crystal structures of the GPCRs, β2AR bound with epinephrine (4LDO), 5-HT_2B_ serotonin receptor bound with lysergic acid diethylamide (LSD) (5TVN), and β1 adrenergic receptor (β1AR) bound with 4-indole piperazine (3ZPQ), share conserved serine residues, 5.42, 5.43, and 5.46, respectively. The 5-HT_2B_ crystal structure and MD simulations show that extracellular loop 2 (EL2) plays the role of a lid at the entrance to the binding pocket, slowing LSD’s binding kinetics, which can be seen in Figure 4 [78].

There are conserved hydrophobic residues located in EL2, such as isoleucine 184 (I184^EL2^) in D2R, which may lead to β-arrestin signaling in this receptor. McCorvy et al. also did biophysical experiments to test the structure-functional selectivity relationships of the designed compounds such as indole N-substitutions. They were intended to disrupt interactions with TM5 and confirmed the validity of their simulations. MD simulations used a homology model based on the dopamine D3 receptor with the antagonist eticlopride (3PBL). MD simulations were used to predict EL2 engagement and find the key to biased signaling. The designed compounds contain an indole-piperazine moiety. The MD package AMBER14 was used. MD simulation results show that the conserved D114^3.32^ forms a salt bridge with the protonated nitrogen piperazine ring in the orthosteric site. To better understand the mechanism between G protein activation and β-arrestin signaling, MD simulations were done without dihydroquinoline-2-one and the alkyl linker. These simulations showed that in the head group of Compound 1 (Figure 5a), the hydrogen on the indole formed a hydrogen bond with S193^5.42^ and did not interact with I184^EL2^. Instead of forming a hydrogen bond, the nitrogen on the indole of Compound 2 was tracked by S193^5.42^; the center of the indole is closer to I184^EL2^ than Compound 1. The two compounds show different poses and interactions with GPCRs. Compound 1 is associated with G protein activation, and Compound 2 (Figure 5b) is associated with β-arrestin biased signaling. Based on these critical simulation results, the authors were able to design new arrestin-biased compounds [76].

### 2.3. Identification of New Ligand Binding Sites and Activation Mechanisms by Accelerated Molecular Dynamics and Metadynamics Simulation

MD simulations have been commonly used to study GPCR activation mechanisms and conformational changes [79,80,81]. However, in the study of conformational dynamics, sampling the extended time-scales involved is the greatest challenge for MD simulation studies. To overcome this challenge, computational scientists have invented new methods and algorithms. Continued increases in computing power have also been of assistance. Currently, commonly used advanced hardware includes powerful graphics processing units (GPUs), supercomputers, and cloud computing [39,82]. Conventional molecular dynamics (cMD) simulations have been widely used to study GPCR activation mechanisms. cMD can reach timescales of microseconds [83,84,85,86,87,88]. However, there are many cases of GPCR activation processes that can take longer than the timescale limits of cMD simulation. Two of the most popular computational methods for enhanced sampling of protein molecular dynamics simulations to access longer time scales are accelerated molecular dynamics (aMD) and metadynamics [89]. aMD simulation improves conformational space sampling by adding a bias potential into cMD that reduces energy barriers between different states [90]. Metadynamics simulation parameterizes the model system with collective variables and introduces bias potentials to discourage resampling of explored conformational space [91]. As a result, both aMD and metadynamics simulations reduce calculation time and are much faster than regular cMD simulations at the risk of utilizing modified energy landscapes. Here, we discuss an example of using cMD and aMD simulations to find a new ligand-binding site and activation mechanism. P2Y_1_R is a purinergic GPCR that is activated by adenosine 5′-diphosphate (ADP). It plays an important role in platelet aggregation and thrombosis formation [92,93]. The crystal structure of P2Y_1_R bound with the antagonist MRS2500 is available on the Protein Data Bank (4XNW); however, there is no structure of P2Y_1_R bound with an agonist. Li et al. used cMD and aMD simulations to find a new agonist-binding site [94]. The crystal structure of P2Y_1_R bound with MRS2500 (4XNW) was used for simulation. The agonist, 2MeSADP, was docked to the same site as MRS2500. The results showed that the aromatic adenine ring of 2MeSADP has a π-π stacking interaction with Tyr303^7.32^. The pyrophosphates interact with Arg128^3.29^ and Arg310^7.39^. However, these results did not match the experimental results. The experimental results show that the mutagenesis of His132^3.33^, Tyr136^3.37^, and Lys280^6.55^ decreased the binding affinity of 2MeSADP with P2Y_1_R. So, aMD simulations were used to run the long time-scale simulations necessary to find an alternative binding site, which can be seen in Figure 6. In this research, all MD simulations used the particle mesh ewald molecular dynamics (PMEMD) module from AMBER14. The ff99SB force field was used for the receptor, and the general AMBER force field (GAFF) was used for the ligands.

The aMD simulations results show the aromatic adenine ring of 2MeSADP interacting with His132^3.33^ through π-π stacking interactions. The N^1^ in the adenine forms hydrogen bonds with the Tyr136^3.37^ and Thr221^5.42^. The pyrophosphates interact with Arg128^3.29^, Arg287^6.62^, Arg310^7.39^, Lys280^6.55^, and Tyr306^7.35^. These results match the experimental results. In this research, metadynamics simulations were used to obtain the potential of mean force for helices III-helix VI, which showed that 2MeSADP-P2Y_1_R has three states: inactive, intermediate, and active. Apo-P2Y_1_R and MRS2500-P2Y_1_R only have two states. The aMD simulation allowed 2MeSADP-P2Y_1_R to pass through the intermediate state and finally reach the active states from the inactive state [94].

### 2.4. Study of Allosteric Modulation and Dynamics of GPCR-Ligand Binding

Traditionally, research has focused on the orthosteric binding sites of GPCRs. There has been growing recent interest in allosteric sites, especially regarding their advantages for drug discovery. Allosteric ligands can act as positive allosteric modulators, negative allosteric modulators, or neutral allosteric ligands. One of the advantages of allosteric modulators is that they have potentially better selectivity than orthosteric ligands [95].

In studying the mechanisms of allosteric modulators, experiments such as x-ray crystallography, NMR, and systematic mutagenesis experiments have had important roles, and these techniques have already given researchers much conformational information of GPCRs and GPCR-mediated signaling. However, these methods only provide information on static conformational states, with the X-ray crystallography experiments only resolving two conformational states, active and inactive. There is no information about the conformations and transitions between these two states. The transition between inactive and active states is difficult to access by experimental methods. Currently, a more intricate, multi-state model, is being investigated by researchers [9,96]. Molecular dynamics simulations give the possibility of revealing the intermediates of GPCRs between multiple states [9,97,98]. In the following section, we will discuss how molecular dynamics simulations provide information on intermediate states and can also identify new allosteric binding sites.

The muscarinic receptors are drug targets for many diseases such as overactive bladder, chronic obstructive pulmonary disease, and neurodegenerative diseases [99,100]. Drug discovery for the muscarinic receptors has been hindered due to the challenge of selectively targeting receptor subtypes. The crystal structures of muscarinic M3 and M4 receptors show the antagonist tiotropium (TTP) in the orthosteric site (PDB: 4DAJ, and 5DSG). The crystal structures show that the orthosteric binding site is highly conserved [101,102]. However, there is no crystal structure of a muscarinic receptor with agonist available yet. Chan et al. used MD simulations to simulate acetylcholine (ACh), the endogenous agonist, binding to M3 and M4 receptors [103]. Figure 7 shows the chemical structures of ACh and TTP. The simulation results show ACh binding to a new, deeper allosteric site near D^2.50^ that is highly conserved [69]. To further understand the ACh activation process, an all-atom MD simulation was used to determine the ligand entrance pathway. ACh binds the orthosteric site, an aromatic cage, quickly (0.1–0.2 μs). ACh was superimposed over TTP in the M3-TPP structure; both of their conformations are the same. After 0.5–0.6 μs into the simulation, the ACh goes deeper into a new binding site, and the pocket size is expanded. In contrast, during the entire simulation, TTP stays at the orthosteric site, as can be seen in Figure 8.

The simulation also shows ACh interacting with residues A112^2.57^, I116^2.53^, A150^3.35^, S154^3.39^, and W503^6.48^. In the M4 receptor MD simulation, ACh shows a similar binding behavior with the M3 receptor, with ACh getting into a deeper binding site next to D78^2.50^. The only difference is that there is an ionic interaction between the quaternary nitrogen of ACh and D113^2.50^. In the M4-ACh model, the quaternary nitrogen interacts with I81^2.53^, V115^3.35^, S116^3.36^, and S119^3.39^ because the ACh flips by 180 degrees.

MD simulations are also used to determine free energies. From the extracellular surface to the orthosteric site to the new binding allosteric site, there are energy barriers between the orthosteric site and the new binding site. This explains why ACh slowly moves from the orthosteric to the new binding sites. These results demonstrate that molecular simulations can be used to determine free energy differences between different binding states, which means that it can also potentially be used to design new allosteric ligands.

In this research, the CHARRMM36 force field was used for the receptor, and the CHARMM CgenFF force field was used for the ligands. The MD package GROMACS was used. The simulation was run longer than 3 μs. Hence, these MD simulations have allowed researchers to see the entire process of ACh binding to the orthosteric and allosteric sites. They also provided unique insights into mechanistic differences induced by the agonist and antagonist. Both TTP and ACh have a quaternary nitrogen. The simulation results suggest that the positively charged nitrogen can be stabilized by forming ionic interactions with the highly conserved Asp^2.50^ in both M3 and M4. This may help researchers identify and optimize future drug candidates. The successful identification of specific allosteric modulators for muscarinic receptors has already inspired similar approaches targeting other receptors involved in neurodegenerative and psychiatric diseases [104].

An interesting interaction between a positively charged ligand nitrogen atom and a negatively charged receptor aspartate is also observed in the simulation studies of μOR. The two opioid ligands, fentanyl and morphine, both have a protonated nitrogen amine in the piperidine ring. Both interact with Asp147^3.32^. In M3 and M4, the positive nitrogen charges form interactions with Asp^2.50^. The positive nitrogen charges in the ligands apparently play a central role in binding and activation.

## 3. Insights of Molecular Dynamics to Drug Design

Drug discovery is among the most challenging of scientific enterprises: it is high risk, high cost, and requires a long time to move from the bench to the market [105]. Early-stage structure-based drug design has the potential to de-risk and accelerate projects. The recent, tremendous advances in GPCR crystallography have provided new opportunities for structure-based drug design [106,107,108,109]. Increasing experimental evidence shows that the GPCR-signaling pathway is more complicated than classical signaling [11]. The multiple mechanisms of GPCR activation and regulation offer diverse possibilities for drug discovery [110,111].

GPCRs undergo major conformational changes during their functional cycles. Researchers frequently want to design a ligand that can specifically bind to the target, in addition to wanting the ligand to activate or inactivate a desired signaling pathway with little to no off-target effects. To reach this point, the designed ligand needs to bind to a specific binding site so that it induces certain conformational states of the receptor. Therefore, understanding the structural dynamics and the mechanisms of various signaling pathways of GPCRs is crucial to the design of GPCR-targeted ligands. As described above, GPCR functional mechanisms are extremely complicated. Hence, understanding receptor dynamics is very important for drug design. There is growing interest in using allosteric modulators for GPCR drug discovery [112]. It has been difficult thus far to identify new allosteric binding sites from crystal structures, in the rare cases that crystal structures are even available. MD simulations can help researchers solve these problems.

Enhanced sampling methods play an important role in this research. Adaptive biased techniques offer the chance to shorten simulation time, thus decreasing computational cost, and let ligands explore more conformational space by making it easier to escape local minima. aMD, one of the more popular adaptive-biased methods, allows the dynamic processes that are required to have agonists pass some energy barriers and reach new energy states. This should allow the possibility of finding new allosteric sites. Metadynamics allows researchers to observe the two-dimensional energy landscape, thus easily seeing the energy differences in all the states. They can serve as a reliability check on the results that aMD provides [94]. To achieve these enhanced sampling methods, the most popular MD packages currently available include AMBER, NAMD, GROMACS, and CHARMM. To obtain an accurate result, the force field chosen plays a very important role. The ff99SB and ff14SB force fields are those most commonly used with AMBER. ff14SB is similar to ff99SB; however, ff14SB was modified by empirical adjustments of the protein backbone dihedral parameters, ϕ and ψ [113]. CHARMM36 is a force field commonly used with NAMD and CHARMM, which includes improved refined backbone CMAP (a grid-based correction for the φ-, ψ-angular dependence of the energy) potentials and side-chain dihedral parameters [114]. To obtain new and accurate insights from simulations, the inclusion of lipids in the model system is important. Experimental results show that bovine rhodopsin is sensitive to lipid environments [115,116]. Crozier et al. are the first to report computational insight about rhodopsin-lipid interactions [117]. They found differences of lipid accessibility differences for the transmembrane helices of rhodopsin. Later, Lyman et al. reported the adenosine A_2A_ receptor in a cholesterol-free POPC membrane, both with and without the antagonist ZM241385. It showed that there is a gap between TM1 and TM2, which allows the lipid headgroup into the binding site and causes receptor instability when there is no ligand present. This prediction was proved later [118]. This is also a good example of long-time scale simulation, with the time scale being 3 μs. Simulations with longer time scales may give us new insights on slower developing phenomena. Previously, simulations performed by Lipinski et al. were longer than 1.2 μs, from which the “Trp rotamer toggle switch” was found.

The primary goal when designing a GPCR-targeted drug is normally to make a ligand that, in addition to binding the desired target, also creates a specific signaling profile. To achieve the desired signaling profile, the drug needs to be able to stabilize certain conformational states of the receptor. If the desired conformational change involves an agonist creating more stabilized active states compared to inactive states, something that needs to be taken into account is how minor changes in the structure of the binding pocket can be associated with different signaling profiles and different intracellular coupling interface conformations [9]. There are challenges involved in successfully doing this. Small changes in one area of the binding pocket can have larger effects elsewhere in the pocket. Changes in an area completely outside of the pocket itself can also have small or large effects throughout the pocket, and vice versa; the latter of which can be important when an allosteric modulator is introduced after the orthosteric ligand is already in the orthosteric binding site. Further difficulties can arise if one seeks to design a modulator that only affects arrestin signaling when changes to any of the sites mentioned above can have an effect on G protein and/or arrestin signaling [9].

The potential contributions of simulations include helping identify important interactions the orthosteric ligand can make with the binding pocket or rearrangements of the binding pocket induced by the ligand. They can also assist with characterizing receptor pocket dynamics, both for known structures based on experiments as well as intermediate or metastable states that are difficult or impossible to currently access experimentally. Another potential use is assessing how ligands affect the pocket and receptor dynamics, and comparing dynamics of closely related GPCRs, which should allow more precise and specific ligand design [9].

A simulation-based approach was used to design chemical modifications that substantially altered a modulator’s allosteric effects on the M2 muscarinic receptor. The modulators the researchers initially studied all partially block the ingress and egress of orthosteric ligands. The allosteric binding site is along the path that the orthosteric ligands take to bind to the orthosteric site, which is curious because the modulators studied also weakened the association and dissociation of the orthosteric ligands. Simulations indicated that the ligand interaction mode was different than initially proposed. The researchers were able to design new modulators that took advantage of information about ligand interactions from the simulation results; the measured affinities of the new modulators were consistent with those predicted by the simulations [43].

## 4. Summary and Perspectives

Computational methods are essential tools for biomacromolecular structural studies. GPCRs are the largest class of drug targets and have structural flexibility, dynamic structures, and complex biological functions. Recent breakthroughs in GPCR crystallography have enabled accurate and predictive MD simulations. Here, we have reviewed recent works that have used MD simulations and enhanced sampling methods to study interactions with new ligands, characterize unknown active/inactive states, and identify new binding sites. This has allowed researchers to gain insights to study new potential drug candidates and obtain qualitative structural information in less time. With computers and algorithms continually growing faster, computational methods will be even more effective in helping future researchers reveal the inner mysteries of GPCRs and their ligands.

The ever-increasing number of GPCR structures found by molecular dynamics simulations of crystal structures will provide a growing database from which new ligands and potentially new binding sites can be determined and explored. Because testing in silico is less resource intensive than in vitro or in vivo, this should allow researchers to find new interesting drug targets.

## Figures and Tables

**Figure 1 ijms-20-04237-f001:**
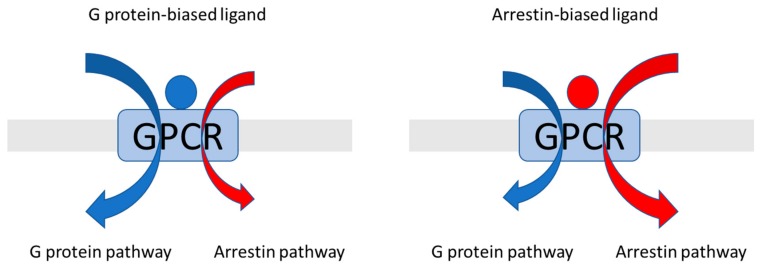
Overview of pathway-biased ligand activation for G protein-coupled receptors (GPCRs).

**Figure 2 ijms-20-04237-f002:**
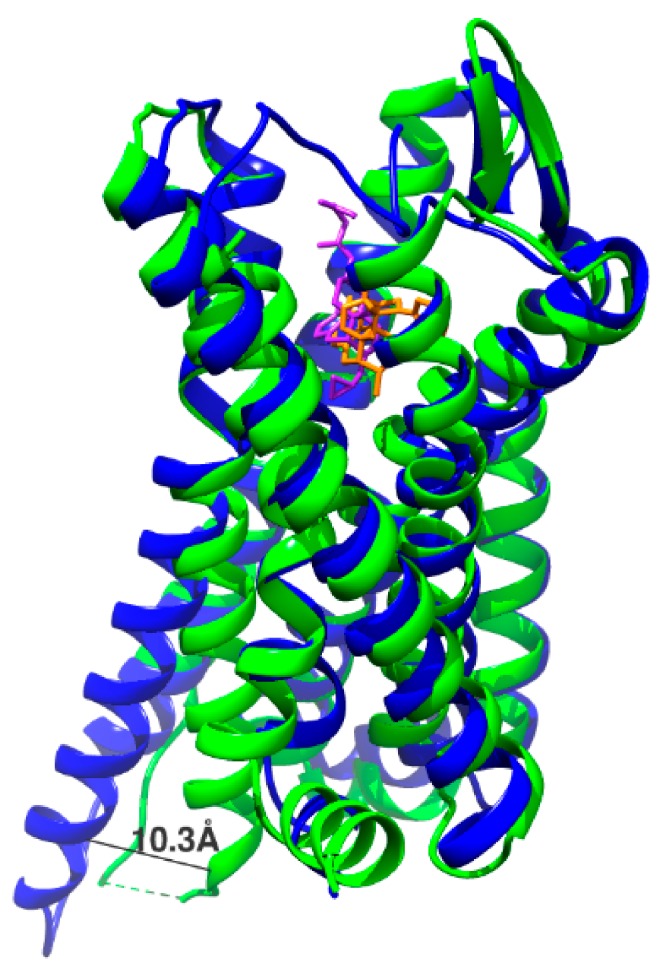
Activation of µOR displaces TM6 by 10.3 Å. The inactive state is in blue and is bound to the antagonist β-FNA (purple). The active state is in green and is bound to the agonist BU72 (orange).

**Figure 3 ijms-20-04237-f003:**
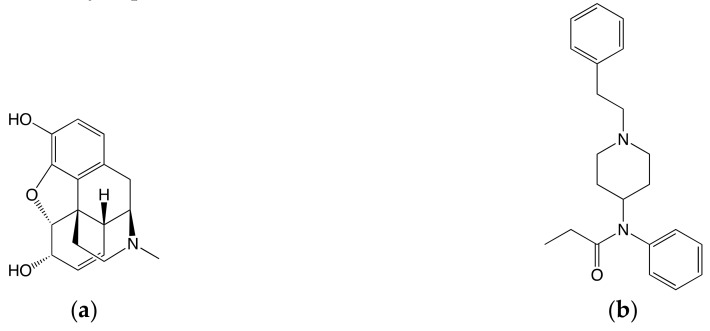
Chemical structures of μOR agonists: (**a**) morphine; (**b**) fentanyl.

**Figure 4 ijms-20-04237-f004:**
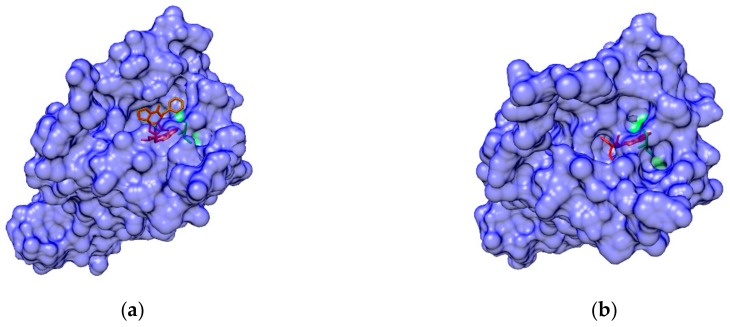
The binding pocket of 5-HT_2B_ is (**a**) open when ergotamine binds and is (**b**) partially closed by movement of L209^EL2^ when LSD binds.

**Figure 5 ijms-20-04237-f005:**
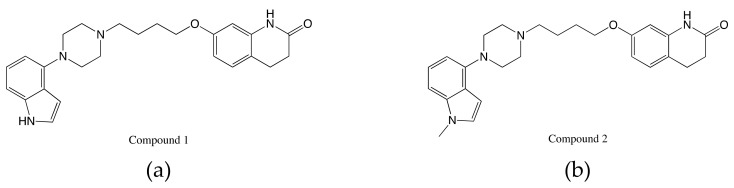
Structures of indole-aripiprazole hybrid compounds used to investigate D2R biased signaling: (**a**) Compound 1; (**b**) Compound 2 [76].

**Figure 6 ijms-20-04237-f006:**
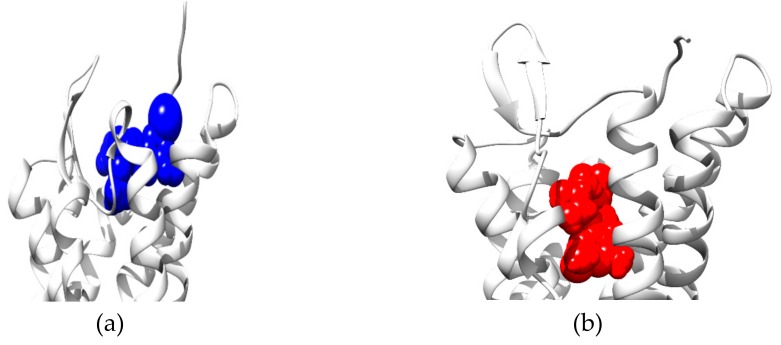
P2Y_1_R binding sites for (**a**) MRS2500 and (**b**) 2MeSADP.

**Figure 7 ijms-20-04237-f007:**
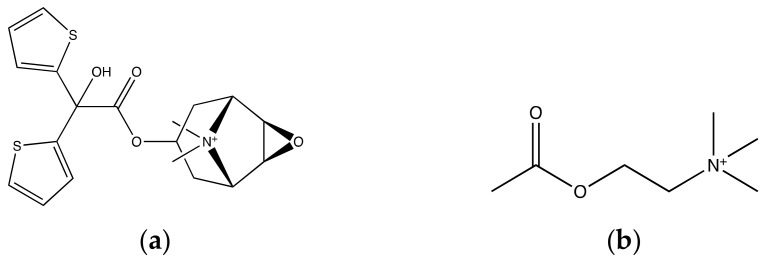
Chemical structures of muscarinic receptor ligands: (**a**) tiotropium (TTP); (**b**) acetylcholine (ACh).

**Figure 8 ijms-20-04237-f008:**
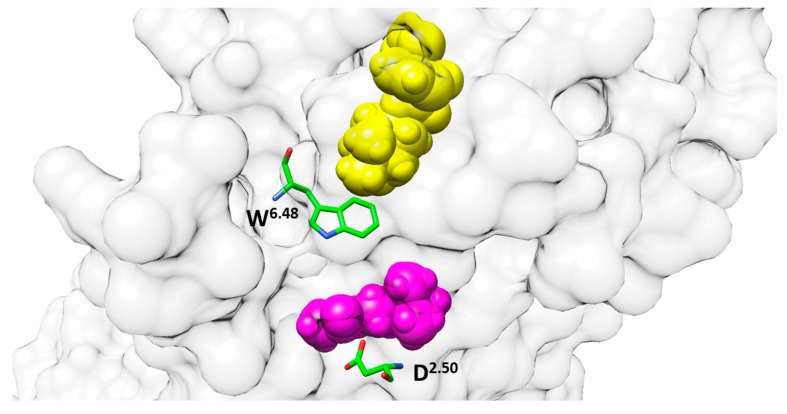
The M3 muscarinic receptor orthosteric binding site is near W^6.48^, whereas the new allosteric binding site is near D^2.50^.

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
