# Peer review of "Recent Insights from Molecular Dynamics Simulations for G Protein-Coupled Receptor Drug Discovery"

_ijms, 2019, doi:10.3390/ijms20174237_

Round 1

Reviewer 1 Report

The ms. by Ng and co-workers reviews the recent applications
of molecular dynamics simulations to GPCR interaction with drugs.
It provides a useful reference for researchers aiming at applying
MD to the study and the design of ligands interactiong with these
proteins.
For this reason I would find useful to report some schemes about the
proteins (e.g. a cartoon with domains and structural elements) and
a cartoon about the processes described in the introduction.
In general figures illustrating the text (e.g. graphical representation
of binding sites from PDB) would greatly ease reading the manuscript.

Minor issues:

- The numbering of residues 5.42, 5.43, ... reuqires an explicit explanation
- p.5 line 220 a comma is missing between inactive and intermediate.
- p.6 line 262 free energies between -> free energy differences between

Author Response

We thank the Reviewer for his/her suggestions. We have corrected the errors. We have also added an explanation for the Ballesteros-Weinstein numbering system for GPCRs. We have also added several new figures to illustrate the structural details discussed in the text.

Reviewer 2 Report

The review by Zou Y. et al. titled "Insights from molecular dynamics simulations for GPCR drug discovery" describes the general current knowledge of the of the molecular dynamics simulations applied to GPCR focusing on the biased-signaling pathway and B-arrestin.

I feel that the review could substantially benefit from some significant revisions. 

The authors should undertake some more insightful discussions. They partly did this but it could be remarkably improved. 

Besides the paragraph 3 "Insights of Moelcular Dynamics..." it would be very appreciated if also in the previous section the authors could shorten little bit the molecular details and emphasize little bit more the discussion.

Eventually, it would be of enormous utility add a figure depicting the intermolecular contacts between the a GPCR and its cognate ligand for each of the examples that the authors analyse. If it is not possible for each at least for most/some. The authors are strongly encouraged to fulfill this request. It will be extremely helpful for the readers.

Apart these major concerns there are also some minor comments that are below shortly listed:

1) the references could be trimmed because some look doubles;

2) it would be strongly appreciated if the authors could explain little bit more in detail the mechanisms of GPCR biased activation;

3) some details must be edited e.g. in the first sentence of the section "Introduction" (pag. 1 line     2) GPCRs they are not "a large superfamily of membrane receptors" as the authors state, but the GPCR family represents the largest family of cell-surface molecules involved in signal transmission!

4) please clarify what does the superscript, beside the aminoacid, indicate. Rephrase sentence page page 4 line 164 "...share conserved serine residues, 5.42, 5.43, and 5.46". What does stand for serine residues 5.42, 5.43 
5) few typo mistakes need to be edited

6) ref. #103 does not fit at all!!! remove or replace

7) minor spell check required

Author Response

We thank the Reviewer for his/her suggestions. We have corrected the errors in the text and references. We have also added an explanation for the Ballesteros-Weinstein numbering system for GPCRs. We have also added several new figures to illustrate the structural details discussed in the text. We have added more background and discussion regarding applications of the discussed research to drug discovery.

Reviewer 3 Report

The publication is a review and the topic suggests a broad aspect of the application of molecular dynamics in drug design. The subject matter is interesting and modern, however, the nature of the publication is a bit chaotic. The authors suggest that the publication focuses on the subject of biased activation (line 56), but in reality the work deals with a few different aspects.

The subject of the application of molecular dynamics in the pharmacological aspect is very wide and in recent years the number of publications on this subject is huge one, it seems that the authors have treated this topic exceptionally selectively and focused on a few publications whose results have been presented.
In my opinion, the authors should develop the subject matter, analyze a broader range of publications on this subject or narrow down the scope of this work.

IIn my opinion, in this type of publication it would be worth to also briefly present the basic methodological aspects of molecular dynamics, which will expand the educational aspect of this publication

Author Response

We thank the Reviewer for his/her comments and suggestions. We have added several new figures to better illustrate the structural details discussed in the text. We have added more background and discussion regarding applications of the discussed research to drug discovery. As the Reviewer points out, there have been many papers covering structure-based drug discovery for GPCRs recently. Thus, we have focused on a specific aspect, that of molecular dynamics applied to GPCRs, which we feel have not been as well covered. There are also many reviews of MD simulation methodology already, many of which we have cited. We have added additional information and citations of methods such as accelerated MD and metadynamics. We think a lengthier review would be more appropriate for a specialized review journal rather IJMS. Thank you for your thoughtful comments.

Round 2

Reviewer 2 Report

The revised version of the review titled "Insights from molecular dynamics simulations for GPCR drug discovery" by Zou Y. et al. benefited from the introduced adjustments -especially the implementation concerning the figures-. I thank the authors for their efforts aimed to substantially improve the manuscript. I appreciated the job, thus currently I consider the manuscript acceptable for publication in IJMS.

Author Response

Thank you. We have further improved Figs. 6-8 upon the suggestions of the Editor.